# Evaluating the Psychometric Properties of the eHealth Literacy Scale (eHEALS) among Polish Social Media Users

**DOI:** 10.3390/ijerph19074067

**Published:** 2022-03-29

**Authors:** Joanna Burzyńska, Magdalena Rękas, Paweł Januszewicz

**Affiliations:** Institute of Health Sciences, Medical College, Rzeszow University, 35-959 Rzeszów, Poland; mrekas@ur.edu.pl (M.R.); pjanuszewicz@ur.edu.pl (P.J.)

**Keywords:** e-health literacy, eHEALS, social media, online health information, reliability, validity

## Abstract

Social media have become mainstream online tools that allow individuals to connect and share information. Such platforms also influence people’s health behavior in the way they communicate about personal health, treatment, or physicians. Individuals’ ability to find and apply online health information on specific health problems can be measured using a valid and reliable instrument, the eHealth Literacy Scale (eHEALS). The objective of this study was to evaluate the psychometric aspects of the Polish version of this instrument (eHEALS-Pl) among social media users, which has not been explored so far. We examined the translated version of the eHEALS in a representative sample of Polish social media users (n = 1527). CAWI (computer-assisted web interviews) was a method to collect data. The reliability of the eHEALS-Pl was measured by calculating the Cronbach alpha coefficients and analyzing the principal components. Exploratory factor analysis and hypothesis testing was used to assess the construct validity of the instrument. The internal consistency of the eHEALS-Pl was sufficient: Cronbach alpha = 0.84. The item-to-total correlations ranged from *r* = 0.514 to 0.666. EFA revealed a single structure explaining 47.42% of the variance, with high factor loadings of the item ranging from 0.623 to 0.769. Hypothesis testing also supported the validity of eHEALS-Pl. The eHEALS-Pl evaluation supported by social media users reviled its equivalence to the original instrument developed by Norman and Skinner in 2006 and it can be used to measure e-health literacy. Since there is no prior validation of the eHEALS among social media users, these findings may indicate important directions in evaluating digital skills, especially in relation to the current challenges related to the COVID-19 pandemic.

## 1. Introduction

When the first email was sent in 1971, probably no one expected that more than 50 years later, the development of online communication tools would radically change the way people communicate. In the second decade of the 21st century, the Internet optimally ensures the need for intuitive, unlimited access to information, including information related to health. According to Flash Eurobarometer 404, published in 2014, 8 out of 10 Europeans (80%) have used the Internet for private purposes—and the majority of respondents (59%) reported using the Internet to search for health information [1]. More than 88% of Polish Internet users admit that they are looking for health information on the World Wide Web. Most of them believe that this information is useful and apply it in daily life [2]. Based on the national survey conducted in 2013 by the Pew Research Center’s Internet & American Life Project, 1 in 3 American adults said they went online to determine their medical condition, and 35% of the respondents said they do not need a professional opinion from a doctor [3]. According to a 2019 survey, 50% of American adults seek health information from online resources [4].

In Poland, the people who most often use the Internet to obtain information about a healthy lifestyle, as well as diseases and treatments, are young and middle-aged people. For example, 55% of people aged 25–44, 49% aged 17–24, and 42% aged 45–59 used the Internet in 2016 to obtain knowledge and information about diseases and treatments. Among people aged 70 to 79 it was 13, and those aged 80 and older almost 5% [5]. Poles appreciate changes in the healthcare system that increase comfort and improve the availability of services; 55% prioritize health over success, fame, professional work, or prosperity and wealth. In a CBOS survey (Center for Public Opinion Research) from January 2019, health was only surpassed by family happiness (80%), which for years has consistently ranked first place among the most important values that Poles follow in their daily lives [6].

In light of this condition, the dynamics of social media development is also interesting. It is estimated that 1 in 4 people worldwide use at least one of these platforms, with Facebook and WhatsApp invariably popular [7]. This new creation, which takes the form of online communities, offers an opportunity previously unknown to receive information, advice, and support from people who are not professionally related to the healthcare sector. It also opens up new possibilities for the distribution of health information, which has taken on a new online formula, changing the sender distribution environment and communication channel. Due to such dynamic changes in the communication landscape, there is a justified need to ask about the health competence of those who are looking for information about their health online. It is recognized that one of the most pressing challenges of contemporary public health should be to shape the competencies responsible for the ability to critically evaluate health-related information. These competencies were defined as e-health literacy. Specialists are alarmed that excessive trust in the so-called “Dr. Google” and the uncritical use of tips found on the web and unreliable, unverified sources can result in unpredictable effects on health and be life-threatening [8].

According to Norman and Skinner, e-health literacy should be defined as ‘the ability to seek, find, understand, and evaluate health information from electronic sources and apply the knowledge gained to address or solve a health problem’ [9]. The authors of this concept believe that the possession of e-health literacy can be a support for solving medical problems that global users of the World Wide Web face every day. The condition is that they have the ability to perform basic and advanced information retrieval, to distinguish scientific articles, reports, and other documents from authoritative and reliable sources, and to understand the chosen e-health terminology. It is also important to preserve critical thinking about the nature of the media itself and to efficiently navigate a wide range of electronic resources to obtain the information necessary to make decisions related to health [9]. Given the wide range of skills required to obtain them, Canadian researchers have identified six components based on which they created the eHealth literacy model. These are literacy and numeracy skills, health competency, information, scientific, media, and IT competences [9]. Norman and Skinner presented the above skills in the form of the ‘eHealth Literacy Lily Model’, using the lily flower as a metaphor for the 6 ideas of eHealth literacy: traditional literacy and numeracy, media literacy, information literacy, health literacy, computer literacy, and scientific literacy. The instrument measuring these literacies was created by the same Canadian researchers called the eHealth Literacy Scale (eHEALS) [10].

Despite the multiplicity of classification and great interest among foreign researchers, illustrated by many publications in this field, the phenomenon of electronic health literacy appears to be insufficiently identified in Poland and is thought-provoking—undervalued. There is only one Polish study on translating, adapting, and measuring the eHEALS properties [11], and yet, e-health literacy was recognized as one of the key challenges of global public health at the turn of the 20th and into the 21st century [12,13,14]. The level of health literacy has been shown to affect the length of health in life, facilitate the ability to cope with disease, and increase the effectiveness of medical care [15]. Understood as a set of multilevel skills allowing one to freely use the available resources, it seems particularly useful in a situation in which millions of Internet users interested in their health reach for information from sources often lacking an evidence-based nature and burdened with lack of credibility and professionalism, which can result in serious health consequences. Skills that fall within the scope of health literacy appear to also be crucial in the interpretation of specialist content available from many sources, which may be difficult for inexperienced recipients and involve the risk of miscommunication.

In the current study, a psychometric analysis of the Polish version of the e-Health Literacy Scale was carried out, evaluating its reliability and validity among users of social networks who participate in online discussion of health issues.

## 2. Materials and Methods

### 2.1. Study Design

In this study, we translated eHEALS into Polish and tested the psychometric properties of the instrument in both pilot and quantitative studies. The results have been reported according to the Checklist for Reporting Results of Internet e-Surveys [16].

### 2.2. Instrument: eHEALS

The eHealth Literacy Scale (eHEALS) consists of eight items that define the scope of knowledge and trust with respect to perceived skills to find, evaluate, and apply electronic health information to address health-related concerns [10]. E-health literacy is measured on a 5-point scale in the range from 1 (strongly disagree) to 5 (strongly agree). The overall range of the eHEALS score varies from 8 to 40, with a higher score indicating more perceived skills in finding, evaluating, and using electronic information to make health decisions [10]. The original psychometric analysis of eHEALS was tested in an adolescent sample (370 boys; 294 girls; mean age 14.95 ± 1.24). The calculated coefficient alpha was 0.88 and the correlation between the test-retest reliability was 0.68. The item-to-scale correlations ranged from r = 0.51 to 0.76. The validity of the tool was evaluated using factor analysis that yielded a single factor solution (eigenvalue = 4.48, 56% of the variance explained) [10].

Due to the fact that eHEALS is a reliable and easy-to-use instrument to assess the level of perceived e-health literacy used in some surveys abroad [17,18,19,20,21,22], we decided to translate and use it with Polish social media users. It should be noted here that Dutch, Japanese, Spanish, Italian, German, Swedish, Chinese, Arabic, and Iranian researchers have translated this instrument into their own native languages.

Before the validation process, we asked Dr. Cameron D. Norman, one of the authors of the eHEALS instrument, to give his approval to the Polish translation and use of eHEALS in the current study. Such a permission was obtained. The Polish adaptation of eHEALS was developed following the translation and adaptation process of the instrument suggested by the World Health Organization [23] but with some modifications of the procedure. In the first step, two fluent English translators translated the instrument into Polish. The translators were introduced to the purpose of the study, the research procedure, and obtained information about the study sample. Next, both translations were combined to reveal possible significant differences and create a unified Polish version. Semantic or linguistic differences were not revealed at this stage. In the second step of the translation procedure, the instrument was translated back into English and compared to the original eHEALS items. Two translators were again approached, but this time their native language was English. The synthesized translation was compared to the original version of the scale. At this stage, it was found that the translated items were semantically similar to the original ones, therefore, no changes were made to the Polish version (Figure 1).

### 2.3. Content Validity and Pilot Study

A cognitive interview and a pilot study were conducted in May 2019. First, 35 students (20 females and 15 males) from the Institute of Nursing and Health Sciences of the Medical Faculty of the University of Rzeszow were asked to investigate the translated version of eHEALS and performed cognitive interviewing where the structure of the questionnaire was discussed. For clarity, respondents were asked to rate each item for clarity and comprehension using a scale that ranged from 1 (not clear at all/unable to understand at all) to 4 (very clear/easily understood). Feedback on content validity was used for the final version of eHEALS-Pl. Second, we prepared a pilot study with a sample of 155 participants. The main goals of the pilot study were again to verify the understanding of the adopted instrument (eHEALS-Pl) and to optimize the questionnaire, which was intended to be used in the quantitative study. The participants were public health, nutrition, and nursing students: 137 females and 18 males, mean age 21.11 ± 0.37, mostly recruited from cities with less than 10,000 inhabitants, but residents of villages were also represented. Most of the respondents rated their health as good (60%). A total of 25.7% of the participants indicated a very good health status and 14.3% of the respondents rated it as average. No one indicated that their health was worse than average. The pilot study used the initial version of the eHEALS-Pl scale. The results of the pilot study showed that there was good internal consistency for the investment (Cronbach α = 0.80), which allowed us to conclude that the tool was well understood. The scale, in accordance with the theoretical assumption, in the preliminary version showed a one-factor structure in the exploratory factor analysis (EFA).

### 2.4. Recruitment and Participants

The quantitative study was carried out using the computer-assisted web interview (CAWI) technique among social media users. CAWI is a survey method in which respondents fill out questionnaires in the online mode with the use of the communication medium in the form of the Internet network [24]. We chose this research method to reduce the costs and time needed to collect data, eliminate errors at the completion state, increase the anonymity of the respondent, and ensure random selection of participants in relation to the time and location of the survey [24].

To select the study sample, the information about the survey was distributed through the following social media platforms:-Facebook: (1) groups—the information about the survey was sent directly to administrators along with an active link to the electronic questionnaire; (2) fan pages—sending a message to the administrator, with the information about the study along with an active link referring to the electronic questionnaire;-Twitter: by posting information about the survey and marking accounts in tweets with an active link to the questionnaire;-Internet forums: (1) by publishing a post with the information about the conducted study directly in the stream of an active discussion; and/or (2) by sending a message to the forum administrator with a request to inform users about the conducted study;-blogs: through direct contact with authors—a message with information about the survey was sent via a contact form or post in the comments section;-YouTube channels: by publishing the information about the survey along with an active link to the questionnaire in the comments section.

These have been chosen based on the results provided by the Brand24^®^ monitoring tool (https://brand24.com/ accessed on 15 February 2019). The multiphase selection of these specific destinations where the information about the research/invitation to participate was sent, was performed with the use of a random number generator in Microsoft Excel. In preparation for the selection, we focused on the high availability of potential respondents, expecting to reach a diverse group of participants. The approximate size of the study population was found to be close to 50,000 (*N* = 47,946).

Taking into account the above, the qualification to participate in the study was a verified account on at least one social platform. The age of the respondents was not limited. One hundred percent return of questionnaires was ensured according to the settings of the Google Docs electronic questionnaire template. Gender (male, female), age (in years), education (primary and secondary school, basic, high school, postsecondary, bachelor/engineer, master’s degree), place of residence (village, city < 10,000, city 10,000–100,000, city 100,000–500,000, city > 500,000), occupation (student, public sector employee, private sector employee, private entrepreneur, farmer, pensioner, unemployed, other professionally inactive), and perceived health status (unsatisfactory, average, good, very good) of participants were recorded.

Based on the number of questionnaire returns—sample size (*N* = 1527), the estimated maximum sampling error was +/− 2.5% at a confidence level of 95% (*p* = 0.05) for the population size of 50,000 people.

Due to the nature of our study, the anonymity of the respondents was ensured by not collecting or processing the data of the respondents at any stage of the study.

### 2.5. Data Analysis

Descriptive statistics (mean, frequencies, percentages, and standard deviations) of the demographics of the participants and the statistical analysis of the results of the psychometric evaluation of eHEALS-Pl were performed using IBM SPSS Statistics 24.0 (IMB, Armonk, NY, USA). The adopted level of significance was *p* < 0.05.

### 2.6. Reliability

The internal consistency of eHEALS-Pl was assessed with the Cronbach alpha coefficient for the overall scale and each item. Cronbach’s alpha remains the most widely used measure of scale reliability, reflecting the average correlation of items within the scale [25]. According to Nunnally’s recommendations, a value of 0.70 or higher was considered acceptable [26].

### 2.7. Construct Validity—Exploratory Factor Analysis

The construct validity of eHEALS-Pl was first examined using EFA and principal component analysis (PCA) followed by a direct oblimin rotation. Factor loadings greater than 0.71 were considered excellent, 0.63 very good, and 0.55 good [25]. The adequacy of the sample was evaluated with the Kaiser-Meyer-Olkin value (KMO) (expected > 0.07) and the Bartlett’s sphericity test (should be significant) [27]. A scree plot was used to help determine the number of factors to be retained.

## 3. Results

### 3.1. Participants’ Characteristics

A total of 1527 Polish social media users participated in the study with a higher proportion of females—89.8% (*N* = 1371)—and 10.2% (*N* = 156) males. The mean age of the respondents was 32 ± 10.37 years, ranging from 14 to 72 years. The largest fraction of the participants (38.8%; *N* = 593) lived in medium-sized cities (10,000–100,000 inhabitants), and 3.8% (*N* = 58) of the respondents indicated a village as their place of residence. Most of the respondents (75.2%; *N* = 1148) had graduated from university: at the bachelor/engineer level were 37.5% (*N* = 572) and at the master’s level 37.7% (*N* = 576). The largest group of study participants (33.3%; *N* = 508) were public sector employees. Every third respondent (30.6%; *N* = 467) was a student. A relatively large group of private sector employees was also recruited, 317 (20.8%). The study included 96 (6.3%) respondents with unemployed status and 38 (2.5%) disabled pensioners. Participants were also asked for a current assessment of their health condition. The respondents most often rated it good or very good—921 (60.3%)—but less than a third (*N* = 438; 28.7%) assessed their health as average, and 168 (11%) of the respondents described it as unsatisfactory, so it seems that a diverse group of respondents was collected in this study. The vast majority of the respondents declared that they use social media every day or several times a day (*N* = 1396; 91.4%), 6.7% of the respondents (*N* = 102) stated that they visit social networks 5–6 times a week, and 1% of the respondents (*N* = 16) declared that they visit social media 2–4 times a week. Ten respondents (0.7%) visited social networks once a week, and three (0.2%) less frequently than once a week. Finally, 61.4% (*N* = 927) of the respondents stated that social media were useful for finding health information.

### 3.2. Reliability

The internal consistency of eHEALS-Pl was found to be Cronbach’s alpha = 0.84 and met a satisfactory internal consistency criterion. Statistics after excluding one of the eight items did not indicate an increase in reliability: the values of the Cronbach’s alpha calculated ranged from 0.81 to 0.83 (Table 1).

The mean total score of eHEALS-Pl for the evaluated population was found to be 30.69 ± 4.25. The mean score for the response to each item reveals that in seven positions the average values were below 4.0, except for item 7 (4.0 ± 0.677). The correlation of individual items with the total score of eHEALS-Pl ranged from 0.506 to 0.666 (Table 1).

The data collected revealed the existence of a significant disproportion of respondents in each age category, as well as by gender. Therefore, we check the Cronbach’s alpha for both variables. In both data categories the Cronbach’s degree of reliability was found to be relatively high in the range of 0.79 to 0.87 (Table 2).

Furthermore, after deleting the relevant items, the scale does not gain or lose reliability. Table 3 illustrates the relevant statistics in this regard.

### 3.3. Construct Validity—Exploratory Factor Analysis

For factor analysis, the significant findings of the Bartlett sphericity test (χ^2^ [28] = 3822.822; *p* < 0.001) supported the factorability of the correlation matrix, and the high value of the KMO test (0.874) showed adequate sampling.

On the basis of the original structure of eHEALS, one factor was also assumed in our study. Analysis of the main components confirmed this assumption—a single factor was retained based on an initial eigenvalue of 3.79 accounting for 47.42% of the variance explained. All items loaded high on this factor, ranging from 0.623 to 0.769 (Table 4).

The single factor structure of eHEALS-Pl has also been empirically confirmed on the screen plot (Figure 2).

### 3.4. Construct Validity—Hypothesis Testing

The results of the hypothesis testing further supported the construct validity of eHEALS-Pl, as evidenced by a significant association between e-health literacy and general usage of social media and certain platforms. Participants who used social media more frequently have been shown to have a significantly higher level of knowledge of e-health literacy (ρ = 0.141). Similarly, a significant positive correlation was observed with the eHEALS-Pl score in users who declared a more frequent use of Facebook (ρ = 0.069), Instagram (ρ = 0.092), blogs (ρ = 0.243), and online forums (ρ = 0.187). However, the significant negative correlation informs us that active Snapchat users are characterized by a low level of e-health competences (ρ = −0.123) (Table 5).

### 3.5. Interpretation of eHEALS-Pl Scores

The eHEALS-Pl scale was composed of 8 items that examine the perceived level of e-health literacy. In this study, we used a five-point response scale that ranges from 1 (strongly disagree) to 5 (strongly agree), with total e-health literacy scores ranging from 8 (lowest possible e-health literacy) to 40 (highest possible eHealth literacy).

To determine low and high results, we used the median value. The median score on the scale was 31.00, which is close to the arithmetic mean (M = 30.69). Therefore, we divided the respondents into two groups: those with a low e-health literacy score (median ≤31.00) and those with a high score—when the total value is greater than 31. In our study, 776 (50.8%) respondents obtained a low eHEALS-Pl score and 751 (49.2%) a high eHEALS-Pl score.

After analyzing the relationships between the scale items and its overall result (Pearson’s r correlation test), it was shown that all items strongly correlate with the overall scale score, which means that the overall eHEALS-Pl score largely predicts how participants respond to particular items, as well as it being possible to predict the overall score based on the response to the items given (Table 6).

## 4. Discussion

The purpose of this study was to translate a full version of the eHealth Literacy Scale into Polish and to examine its psychometric aspects among social media users. In general, the results of the psychometric analysis showed that the instrument is a universal measurement tool aimed at Internet users, both women and men, in various age categories. Furthermore, the eHEALS scores supported its reliability as an easy-to-use instrument for assessing the level of perceived e-health literacy in different cultural contexts in various countries such as: The Netherlands [17], Japan [18], Spain [19], Italy [20,29], Germany [21], Iran [22], Portugal [30], South Korea [31], and China [32]. Researchers have translated the instrument and used it in a variety of populations, including: schoolchildren [33], adolescents [34], university students [19,21], adults of a wide age range [17,20,35], and older adults [35,36,37]. To broaden the research perspective, we used a sample from the general population, compared to other validations conducted in younger samples [10,33,34] or older samples [35,36,37].

Although a more extensive psychometric analysis is necessary to establish instrument validity, the results of the current study are promising and showed that eHEALS-Pl is a reliable and consistent measurement tool for perceived measurement of e-health literacy.

The exploratory factor analysis revealed that the Polish version of eHEALS has a structure of one factor explaining 47.42% of the variance and that its structure is consistent and has high internal compatibility (Cronbach’s alpha = 0.84). The selected scale items successfully examine the level of perceived e-health literacy, and the sum of the eight variables determines the degree of intensity.

The analyses carried out in the current study do not differ significantly from the original analyses carried out by Norman and Skinner, who revealed a single structure and reliability at Cronbach’s alpha = 0.88 [10]. In the first Polish adaptation of eHEALS, a single factor structure was also reported [11]. In the Dutch version of the scale, the single dimension and internal coherence were shown by Cronbach’s α = 0.93 in study 1 and α = 0.92 in study 2 [17]. Our findings also support previous studies testing the internal structure of eHEALS using EFA, where the scale appears to have a single factor solution explained: 59.72% of the variance in the scale for the sample of older adults and 55.06% of variance for the sample of 18–35-year-old women [11], 52.55% of variance in the Spanish version of the scale (Cronbach’s α = 0.87) [19], 70.5% of variance in the measure in the Iranian version (Cronbach’s α = 0.88) [22], 67% of variance in the study of de Caro et al. (Cronbach’s α = 0.87) [29], 50.3% of variance in the measure in the study of Koo et al. (Cronbach’s α = 0.87) [33], 67.3% of variance found by Chung and Nahm [37], and 59% of the variance explained in the study of Neter and Brainin [38]. There are also promising results in different cultural validations of eHEALS where the scale yielded a two-factor structure [20,21] with the use of an alternative analysis technique: a confirmatory factor analysis (CFA). Diviani and colleagues [20] as well as Soellner and colleagues [21], for instance, used CFA to compare the one-factor model based on the original eHEALS analyzed with the two-factor specified a priori. The results, contrary to our study, indicated a better fit for the two-factor structure in both cultural validations. However, parametric and nonparametric item response theory analyses conducted in the Italian validation of eHEALS confirmed that the single-structure model best fits the data in this particular study sample (adults with mean age 37.37 years, SD 13.78) [20]. Recently, one study which examined the measurement properties of the instrument with the use of CFA provided a good-fitting model comprising a single factor [18], the same as eHEALS-Pl. eHEALS was originally developed in English, and multiple studies with English-speaking samples tested its structure with CFA. Sudbury-Riley et al. [36], for instance, found a three-factor structure of the scale, based on data from three different nations and random samples. The three-factor structure of eHEALS was also supported in an Australian outpatients sample [39] and administered to older adults aged 50 years or older over the telephone [36]. To conclude, the findings of various studies conducted in different countries and samples documented that the internal structure of eHEALS has shown considerable variability. As Stellefson et al. suggested, it could have been affected by the age of the participants [36]. Another reason reported by Sudbury-Riley and colleagues might be related to the translation process and the adaptation of the instrument to specific cultural contexts [35].

In our study, similar to Neter and Brainin [38], we used the median value to determine two groups with a high and low e-health literacy score, as James and Harville also did among African Americans [40] and Wångdahl et al. did in Sweden [41]. A small majority of participants (50.8%) reported a low eHEALS-Pl score and lack of confidence in their skills in the search for online health. Considering that the research was conducted online among active Internet users, this result was a surprise. However, hypothesis testing has shown that participants who used social networks more frequently show a significantly higher level of e-health literacy. Li and Liu reported similar results, which found that e-health literacy (β = 0.07) positively affected the frequency of social media use [42]. At the same time and in the context of a high percentage of trust in online health information and wide access to the Internet among Poles, this aspect of the research conducted encourages reflection and demands to be examined further.

## 5. Limitations

There are some limitations to be noted in this study. First, all participants in our study were Internet users; therefore, this population may not be representative of the general community of Polish citizens. Second, there was a cross-sectional survey, and no test-retest reliability or predictive validity estimates were conducted during this study. Third, it is also important to note that to collect all data, we only used web-based survey methods. Online studies have several advantages, including time and cost efficiency, but are also prone to response bias [22,34]. The fourth limitation refers to the transcultural adaptation process that was carried out without the participation of experts in the field. Although we conducted a two-phase pilot study with two groups of students, most of them were students in public health, nutrition, and nursing, where content validity and cognitive interviewing were performed, so there is a need for a more formal study investigating the final version of eHEALS-Pl. The fifth limitation refers to the data collected in this study. Although our intention was to collect a diverse sample, both in terms of age and education, the distribution of the gender variables was not normal—there was an overrepresentation of females in our study. It could have been due to the fact that women go to health resources on social media more often than men. Therefore, we described the eHEALS-Pl score using the median value. Sixth, we noted the need to verify e-health literacy for people 65+ due to their lower digital competences. It is our suggestion to conduct more in-depth research related to this aspect of e-health literacy.

## 6. Conclusions

The results of the current study confirmed that the items in the Polish version of eHEALS were equivalent to the original instrument developed by Norman and Skinner in 2006. eHEALS-Pl was also shown to be a reliable and valid measure of e-health literacy of Polish-speaking Internet consumers who are active social media users. There is a hope that other Polish researchers who intend to measure e-health literacy among various groups of online users can use the instrument in a translated version. Since there is no prior validation of eHEALS in social media users, these findings may indicate insight for further improvement in the performance of eHEALS items in online settings. It is important to determine the applicability of eHEALS-Pl among groups with chronic diseases or at risk of low digital literacy, especially in the context of the COVID-19 infodemic. Our study also has potential for health care providers and public health professionals for whom the widely available online health information raises the need for active creation of e-health literacy among consumers in the health decision-making process. The results obtained in our study not only enrich the theoretical paradigm of public health management and online health communication but also have practical implications in the COVID-19 infodemic. Improving digital health literacy levels is also essential for future infodemic preparation.

## Figures and Tables

**Figure 1 ijerph-19-04067-f001:**
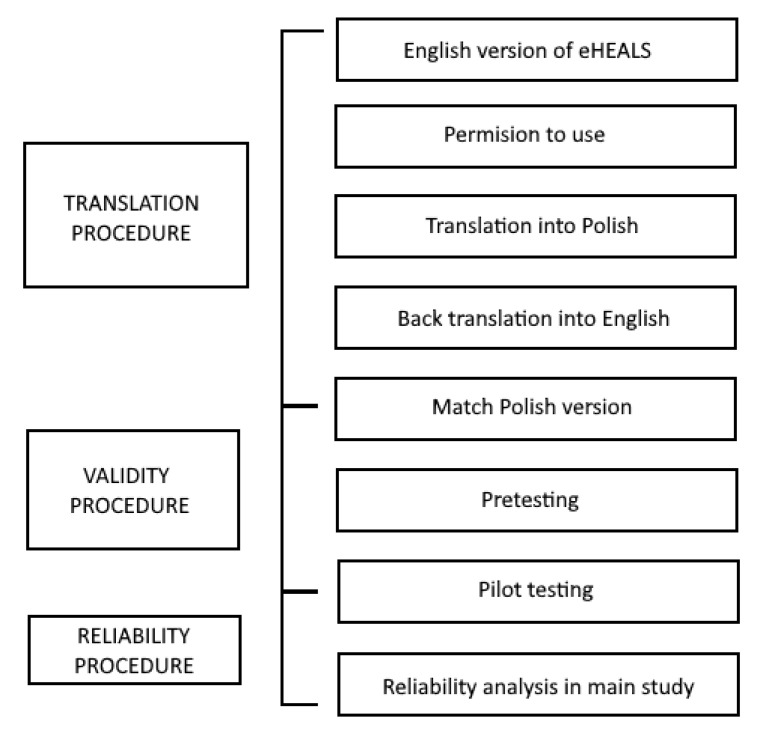
Translational validation of the Pl-eHEALS.

**Figure 2 ijerph-19-04067-f002:**
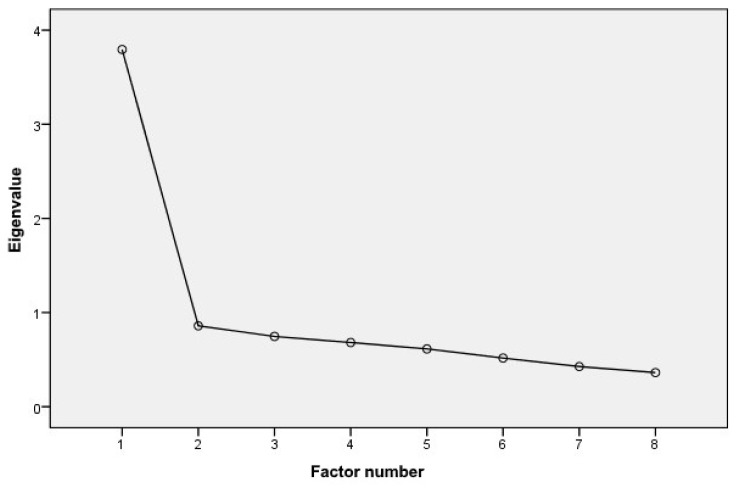
Scree plot for P-eHEALS.

**Table 1 ijerph-19-04067-t001:** eHEALS-Pl means, scale reliability after removing an item, and item-to-total correlation.

eHEALS-Pl Items	Mean (SD ^a^)	Mean, If Item Deleted	α, If Item Deleted	Variance of the Scale, If Item Deleted	Item-to-Total Correlation ^b^
item 1	3.61 (0.946)	27.08	0.814	14.920	0.634
item 2	3.69 (0.985)	27.00	0.809	14.482	0.666
item 3	3.91 (0.731)	26.78	0.829	16.831	0.514
item 4	3.87 (0.798)	26.82	0.815	15.828	0.626
item 5	3.98 (0.756)	26.71	0.830	16.738	0.506
item 6	3.93 (0.722)	26.76	0.828	16.852	0.518
item 7	4.00 (0.677)	26.69	0.829	17.094	0.516
item 8	3.71 (0.906)	26.97	0.817	15.327	0.605
Mean (SD) sum score	30.69 (4.52)				

^a^ SD: standard deviation. ^b^ All item-to-total correlations were significant at *p* < 0.001.

**Table 2 ijerph-19-04067-t002:** Scale reliability correlated with gender and age.

Variable	α	Items
Gender		
female	0.834	8
male	0.869	8
Age (years)		
<17	0.799	8
17–24	0.789	8
25–34	0.808	8
35–44	0.809	8
45–59	0.837	8
60–64	0.807	8
<64	0.847	8

**Table 3 ijerph-19-04067-t003:** Scale reliability correlated with gender and age after removing the item.

Item	Age (Years)							Gender	
	<17	17–24	25–34	35–44	45–59	60–64	64+	Females	Males
item 1	0.777	0.744	0.768	0.793	0.779	0.776	0.804	0.807	0.841
item 2	0.737	0.762	0.764	0.769	0.804	0.805	0.820	0.802	0.843
item 3	0.759	0.770	0.788	0.800	0.784	0.811	0.850	0.821	0.861
item 4	0.745	0.752	0.779	0.780	0.808	0.763	0.818	0.808	0.844
item 5	0.753	0.782	0.792	0.789	0.815	0.751	0.837	0.822	0.864
item 6	0.812	0.780	0.800	0.792	0.857	0.787	0.796	0.822	0.853
item 7	0.836	0.763	0.804	0.796	0.825	0.790	0.857	0.821	0.864
item 8	0.761	0.771	0.785	0.777	0.844	0.794	0.828	0.810	0.851

**Table 4 ijerph-19-04067-t004:** Factor loadings of eHEALS-Pl.

Pl-eHEALS	1
(Q1) I know which health resources are available on the Internet	0.769
(Q2) I know where to find helpful health resources on the Internet	0.741
(Q3) I know how to use the health information I find on the Internet to help me	0.736
(Q4) I know how to find helpful health resources on the Internet	0.721
(Q5) I have the skills I need to evaluate the health resources I find on the Internet	0.634
(Q6) I know how to use the Internet to answer my questions about health	0.633
(Q7) I can perceive which health resources are of high quality and which are of low quality on the Internet	0.633
(Q8) I feel confident in using information from the Internet to make health decisions	0.623
% of variance explained	47.426

**Table 5 ijerph-19-04067-t005:** Spearman correlation between eHEALS-Pl scores and social media use.

*N* = 1527	eHEALS-Pl Overall	
Social media usage—overall	rho	0.141
	*p*	<0.001
Facebook	rho	0.069
	*p*	<0.001
YouTube	rho	0.027
	*p*	0.290
Twitter	rho	0.029
	*p*	0.252
Instagram	rho	0.092
	*p*	<0.001
Snapchat	rho	−0.123
	*p*	<0.001
LinkedIn	rho	−0.29
	*p*	0.250
Wikipedia	rho	0.037
	*p*	0.147
Blogs	rho	0.243
	*p*	<0.001
Online forums	rho	0.187
	*p*	<0.001

rho (ρ)—Spearman rank correlation coefficient.

**Table 6 ijerph-19-04067-t006:** Pearson’s correlations and Spearman’s rho and eHEALS-Pl score overall (*N* = 1527).

eHEALS-Pl Items	eHEALS-Pl Overall	
	*r* ^a^	*ρ* ^b^
(Q1) I know what health resources are available on the Internet	0.750 **	0.719 **
(Q2) I know where to find helpful health resources on the Internet	0.778 **	0.744 **
(Q3) I know how to use the health information I find on the Internet to help me	0.628 **	0.599 **
(Q4) I know how to find helpful health resources on the Internet	0.728 **	0.677 **
(Q5) I have the skills I need to evaluate the health resources I find on the Internet	0.625 **	0.600 **
(Q6) I know how to use the Internet to answer my questions about health	0.630 **	0.591 **
(Q7) I can perceive which health resources are of high quality and which are of low quality on the Internet	0.622 **	0.603 **
(Q8) I feel confident in using information from the Internet to make health decisions	0.725 **	0.691 **

** *p* < 0,0001. ^a^ *r*—Pearson correlation coefficient. ^b^ *ρ* (rho)—Spearman rank correlation coefficient.

## Data Availability

The data presented in this study are available on reasonable request from the corresponding author. The data are not publicly available due to restrictions, e.g., their containing information that could compromise the privacy of research participants.

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
