# Peer review of "Evaluating the Psychometric Properties of the eHealth Literacy Scale (eHEALS) among Polish Social Media Users"

_ijerph, 2022, doi:10.3390/ijerph19074067_

Round 1

Reviewer 1 Report

This study created a Polish translation of eHEALS, surveyed Polish social media users, and evaluated their e-health literacy. This study is interesting in assessing the appropriate access and use of e-health information among the social media users.

However, it is not clear the main finding through this study; Polish version of eHEALS applicability, applicability to social media users, or e-Health literacy among Polish social media users. Because the creation and evaluation of the Polish version of eHEALS has already been reported in Ref. 9, the impact of creating a translated version of eHEALS is small. While the reliability and validity of the eHEALS-PI obtained in this study has been discussed, there has not been sufficient discussion of e-Health literacy among Polish social media users.

Introduction

  • The purpose of this study is unclear. What did this study clarify from the data obtained by eHEALS-PI?

Materials and Methods

  • Was this survey conducted with the approval of the Institutional Review Board?
  • With a huge number of users on each social media, how did you choose your target?
  • Was the target of this survey limited to Polish citizens?
  • 3 L.100 Ref.13 is a typographical error in Ref.14.

Results & Discussion

  • Are there any findings in the mean total eHEALS-Pl score of 30.69 ± 4.25 compared with other previous research?
  • The result of consumers who use social media frequently was high e-health literacy is interesting. Did you evaluate whether the age and/or gender of each social media users was affecting the result?

Conclusion

  • Is the discussion described later in the conclusion based on the results of this study?

Reviewer 2 Report

Some expressions or sentences need to be revised (e.g. lines 75 and 76) and small corrections, especially related to verbal times, should be considered. (e.g. line 53, 55). Sentence 162 to 164 is not clear in its meaning.

Limitations are overall well presented and the authors recognise methods could be improved. A limitation related to the age of the participants should also be referred, even if the distribution by age is acceptable but because older adults are generally less digitally literate and thus a specific  study with ages over 65 would improve accuracy to the results.

Reviewer 3 Report

The presented article presents too many limitations to be used in a generalizable way as a representation of the use of e-health among the Polish population. In addition, the only demographic variables considered are age and sex, not socioeconomic or cultural level, which is determined when analyzing the use of social networks as a source of information, especially in health. In addition, minors are mixed, who attend health services under the responsibility of parents and guardians, as adult personnel.

At a structural level, the theoretical framework is insufficient and little delimited, even, for example, it cites data from US users that has little to do with the Polish population. The conclusions are few and provide little information to the reader.

For all these reasons, the article is not considered acceptable for publication in the current format.

Round 2

Reviewer 1 Report

The manuscript has been improved and I understand the author's comments, I agree to publish it in this revised edition.

Reviewer 3 Report

I agree with the revision made and the improvements introduced. The work may be published in its current form. The deficiencies detected in the previous revision have been greatly improved.